# Wearable Sensor Network for Biomechanical Overload Assessment in Manual Material Handling

**DOI:** 10.3390/s20143877

**Published:** 2020-07-11

**Authors:** Paolo Giannini, Giulia Bassani, Carlo Alberto Avizzano, Alessandro Filippeschi

**Affiliations:** 1TeCIP Institute, Scuola Superiore Sant’Anna, 56127 Pisa, Italy; paolo.giannini@santannapisa.it (P.G.); giulia.bassani@santannapisa.it (G.B.); carloalberto.avizzano@santannapisa.it (C.A.A.); 2Department of Excellence in Robotics and AI, Scuola Superiore Sant’Anna, 56127 Pisa, Italy

**Keywords:** motion tracking, health monitoring, ergonomic assessment, biomechanical overload risk, inertial measurement units, activity segmentation

## Abstract

The assessment of risks due to biomechanical overload in manual material handling is nowadays mainly based on observational methods in which an expert rater visually inspects videos of the working activity. Currently available sensing wearable technologies for motion and muscular activity capture enables to advance the risk assessment by providing reliable, repeatable, and objective measures. However, existing solutions do not address either a full body assessment or the inclusion of measures for the evaluation of the effort. This article proposes a novel system for the assessment of biomechanical overload, capable of covering all areas of ISO 11228, that uses a sensor network composed of inertial measurement units (IMU) and electromyography (EMG) sensors. The proposed method is capable of gathering and processing data from three IMU-based motion capture systems and two EMG capture devices. Data are processed to provide both segmentation of the activity and ergonomic risk score according to the methods reported in the ISO 11228 and the TR 12295. The system has been tested on a challenging outdoor scenario such as lift-on/lift-off of containers on a cargo ship. A comparison of the traditional evaluation method and the proposed one shows the consistency of the proposed system, its time effectiveness, and its potential for deeper analyses that include intra-subject and inter-subjects variability as well as a quantitative biomechanical analysis.

## 1. Introduction

Even in the newest and most automated industrial facilities, manual work still covers a large share of the activities. Workers have an important role to perform tasks when it comes to observation, decision-making, and in other instances, in which tasks benefit from human precision, skill, and movement capabilities. In fact, with the increase of product variants built at the same assembly line, with an associated small order size, human flexibility and cognitive skills are highly needed. Therefore, many workers are still exposed to Work-related Musculoskeletal Disorders (WMSDs). According to the European Agency for Safety and Health at Work, in 2015 58% of workers were affected by musculoskeletal disorders, with higher prevalence in the upper limbs and back [1].

Workers involved in manual material handling (MMH) activities are at greater risk of developing WMSDs [1]. Therefore, the assessment of the biomechanical overload risk related to MMH is a fundamental step in the prevention of WMSDs. The current standard methods and procedures to assess the biomechanical overload risk are mainly observational. An expert rater observes a worker either during the activity or after it by visual inspection of recording and manually fills an appropriate checklist. This checklist allows for the calculation of a risk score and typically requires the measurement of several anatomical variables (e.g., trunk flexion, shoulder abduction, et al.), efforts, and working frequencies. These measurements are typically done qualitatively, being subjective and highly dependent on the rater expertise, and providing unnecessary variability (e.g., Reference [2]. Moreover, this method is extremely time consuming and fatiguing for the rater, which has to keep a high level of attention in the analysis of tens of minutes, or even a few hours of working activity.

With the goals of alleviating the burden for the rater, of making objectives measures available for the score of risk, and to allow for further task investigations based on a biomechanical analysis of the task, this paper presents a novel system for a semi-automatic assessment of the biomechanical overload risk. This system is based only on wearable devices, and it includes both an activity segmentation module and a risk score module that contribute together to give to the rater a score for each phase of the analyzed activity. The paper presents a procedure that allows the rater to use the system from the selection of the activity, through the task analysis, selection of the ergonomic risk assessment methods, and training of the activity segmentation module up to the visualization and storage of the results. Thanks to this system, the rater relies on objective measures and has the opportunity to analyze a much higher number of workers, being able to assess the variability of the risk both within the same worker and among workers. This analysis will provide a much richer representation of the risk of the selected activity, and it will enable to account for both variability and subject-specific performance to decide the risk mitigation actions (see References [3,4,5] for examples of such kind of analysis). As detailed in Section 2.2, the proposed system advances current state-of-the-art methods for MMH risk assessment as it covers the whole body, considers both kinematic and kinetic information, and it embeds an activity segmentation module.

Based on the standard risk evaluation methods listed in the standards ISO1228 and the technical report TR12295 (TR), the proposed method gathers data from the worker employing a wearable sensor network composed of Inertial Measurements Units (IMUs) and surface Electromyography (sEMG) sensors. An analysis pipeline that includes filtering and processing of signal provides kinematic and kinetic variables needed to implement such methods, as well as segmentation of the activities.

The proposed method has been developed in the framework of the Sailport project [6], a trilateral approach to improve both the safety and health of port workers. Port activities are typically unstructured and characterized by high biomechanical overload risks causing a high incidence of Work-related Musculoskeletal Disorders (WMSDs). In particular, the Lift-on/Lift-off of containers provided an ideal context to assess the capabilities of the proposed system.

The paper is organized as follows—Section 2 includes the background, Section 3 describes the proposed system including both its architecture and components, and the procedure that allows the rater to use the proposed system. Section 4 presents the case study that was adopted to assess the system and to give an example of how it can be used. The same section reports the results of this experimental activity and their discussion. Finally, Section 5 concludes the paper.

## 2. Background

### 2.1. Ergonomic Assessment Methods for MMH

WMSDs are often caused by awkward postures, weight transport, and repetitive movements. The ergonomic risk due to biomechanical overload related to such activities is assessed employing an ISO standard, namely the ISO 11228 that subdivides MMH activities into three different sections—ISO 11228-1 refers to lifting and carrying, ISO 11228-2 to pushing and pulling and ISO 11228-3 to the handling of low loads at high frequency. For each section, standard methods are suggested for the evaluation of the ergonomic risk.

This paper refers to four suggested methods—the NIOSH LI (National Institute of Occupational Safety and Health Lifting Index) index for the ISO 11228-1 [7], the Snook & Ciriello method for the ISO 11228-2 [8], and the REBA (Rapid Entire Body Assessment) [9] and the SI (Strain Index) [10] for the ISO 11228-3. Whereas the NIOSH LI and the Snook & Ciriello methods are the golden standards for the respective sections, the methods adopted for the ISO 11228-3 are not the preferred one, and they have been chosen instead of the golden standard method (OCRA) because of the presence of activities with moderate loads and because they are better suited for automatization of the score calculation. Human posture and muscular acivation provide fundamental information for the implementation of NIOSH, Snook & Ciriello, REBA, and Strain Index ergonomic assessment methods. The NIOSH standard requires the computation of the maximum vertical and horizontal displacements, the initial lifting height, and the angular displacement of the trunk. Moreover, the method requires an estimation of the actions’ frequency. All these variables can be automatically determined as described in Section 3.5. Similarly, the Snook & Ciriello method requires the load-displacement and the actions’ frequency. Thanks to the data about reference displacement, hands height, and frequencies obtained from time analyses the Snook & Ciriello methods can be implemented. For an accurate application of the REBA standard, upper and lower body’s anatomical joint angles must be produced to evaluate posture. This standard considers trunk, shoulders, elbows, wrists, head, knees joints. Besides, the frequency of activities, the class of the load applied to the upper limbs, as well as the grip quality needs to be estimated for the evaluation. The strain index is a method of analyzing works for the risk of distal upper limb disorders. This method requires six variables that include an evaluation of the intensity, duration and frequency of effort, as well as the wrist posture and the speed of work. EMG and motion data are used to compute most of these variables.

Section 3.5 reports the details of the calculation of the variables needed to calculate the risk scores according to the aforementioned methods. The same section reports what variables are calculated by the system and what variables are provided by the rater in the task analysis.

### 2.2. State of the Art

Workers’ postural and movement estimations are important elements to determine the risk of musculoskeletal injury in the workplace [11]. Different methods are used for a proper assessment of risk factors exposure for WMSDs [12], namely: self-report methods, observational methods, and direct methods.

Self-report methods consist of various forms such as questionnaires, checklists or interviews, and must be filled by the monitored workers. These methods are easy and quick to use. However, they are not always reliable and could lead to subjective interpretations due to the worker’s perception [13,14].

Observational methods are commonly used in a wide range of ergonomic assessments [12]. Several simple methods have been developed to evaluate worker exposure. Usually, expert operators fill in proforma forms while observing the workers during their activities directly or through video recording observation [15,16]. On one hand, these methods are practical and cheap, on the other hand, this evaluation is questionable for its imprecise nature [12]. Even some advanced observational methods employ the video recording technique to estimate the joint angles. There exist specific software tools that allow for analyzing video recorded manual tasks [17]. However, these methods are problematic to be utilized in work performances due to the possibility of an occlusion, space availability and system complexity [18].

Direct measurement methods base themselves on sensors directly attached to the worker’s body. They include methods based on optical motion capture systems, such as the one described in Reference [19,20] and fully wearable systems. Bortolini et al. [19,20] develops a movement analysis system (MAS) by integrating a network of cameras and software to automatically and quantitatively provide productive and ergonomic information (OWAS, REBA, NIOSH and EAWS ergonomics indices). The fully wearable systems methods allow the analysis of activity in outdoors and unstructured environments while minimizing the influence of measurement activity during worker sessions. Moreover, they allow for real-time postural measurements that enable an online ergonomic evaluation, which, according to Reference [21], has beneficial effects. Indeed, if the system gives to the workers information concerning his/her current ergonomic behavior, they can correct their posture immediately. Vignais et al. [22] propose a real-time RULA assessment based on a direct measurement with a network body inertial sensor and goniometers. Real-time feedbacks help workers to assume the correct posture and, as a consequence, to decrease the risk of developing WMSDs. The system is capable of giving visual feedback only about upper body postural risks, according to what is considered by RULA standard evaluation method. Battini et al. [23] present a real-time system based on whole-body inertial sensors able to integrate several ergonomic assessment methods such as RULA, OWAS OCRA, and Lifting Index (LI). ViveLab Ergo (ViveLab Ergo, Székesfehérvár, Hungary) is commercial software that provides a real-time system based on the IMUs system and implements several ergonomic assessment methods. Direct measurements could also supply information to help activities recognition. Adrien Malaisé et al. [24] show a system that is able to classify different mimicked industrial tasks via the integration of the IMU sensor and e-glove. In recent years, IMU-based motion tracking systems has reached an accuracy in the order of a few degrees for each anatomical joint [25,26]. The accuracy of such systems is sufficiently high to help in the management of musculoskeletal disorders. For example, in Reference [27] an IMU-based system has been compared to an optoelectronic evaluation for whole-body motion analysis, obtaining a non-significant difference between the two methods. Analogously, Hara et al. [28] use inertial units for an automatic assessment of order picking. Other ergonomic assessment methods consider the integration of IMUs sensors and superficial electromyographic (sEMG) measurement sensors that allow the user to measure muscular activity [29,30]. sEMG signals help the ergonomic assessment, in fact, through them, much information could be extracted such as those regarding muscle activation, muscle effort in the different body areas, mean values, peaks, percentiles, cumulative exposure, rate of change. sEMG quantitative muscle effort measurements can be directly adopted to integrate ergonomic standard methods such as SI [29].

This work develops a new full-body semiautomatic real-time system to improve ergonomic risk evaluation. Such a system combines kinematic and EMG signals to perform activities recognition and to provide all the necessary information for the implementation of standard assessment methods. Wearable sensors are taken into consideration for their handiness and possible applications in several workplaces. An internal Neural Network (NN) processes kinematic and sEMG signals for an online worker’s activities recognition. The operator must select only the working state activities so that the aforementioned NN is able to classify such motions and to consequently choose the optimal ergonomic evaluation standard. In this way, the time requested during segmentation and evaluation activity is notably reduced. This system implements four different ergonomic standard methods to cover all ISO 11228 cases. The development of graphic interfaces provides a real-time feedback score to help workers ergonomic behavior assessment. Inertial and EMG sensors allow the obtainment of objective measurements and a greater quantity of data. These data favor both an intra-subject and an inter-subjects analysis of the behavior and the sources of ergonomic risk recorded during the entire working sessions.

The proposed system, compared to the systems listed in the state of the art, combines the movement of the total body, the integration of the EMG signal and the segmentation activity. The major advantage of the full-body system lies in the possibility to implement evaluation methods in which postural information of the whole body is needed, especially the NIOSH and REBA methods, without being limited to particular districts. The measurement of EMG signals allows a more reliable assessment of muscular efforts, its direct application in the evaluation of the Strain Index as well as improving the activity segmentation. In addition, future developments could use the EMG signals for the classification of handled loads. Finally, the possibility of semi-automatic segmentation of activities decreases the time required for manual segmentation.

## 3. Materials and Methods

### 3.1. Architecture

The proposed ergonomic risk assessment system allows for obtaining an ergonomic risk score for different working activities that involve MMH according to the methods reported in Section 2.1. The system, besides the wearable device, involves also the use of a proprietary software implemented on a host PC (Section 3.4). Its architecture is reported in Figure 1 along with the references to the sections that describe each module. The wearable component is composed of the devices that acquire human kinematics and muscular activity data. Kinematic data are gathered through suits composed of IMUs. Three options are available, a custom device [31], and two commercial ones, that is, the XSens MVN (Xsens Technologies, Enschede, Netherlands), and the Perception Neuron (Noitom Ltd., Beijing, China). Muscular activity is recorded using either a custom system or a commercial one, that is, the Shimmer3 EMG (Shimmer Research Ltd., Dublin, Ireland). These devices communicate either via WiFi or Bluetooth with a host PC, that runs several software modules that include task analysis, segmentation, data pre-processing, ergonomic evaluation, and output (see Figure 1). The same figure reports the references to the sections where modules are described.

The task analysis module allows the rater to provide a template of the phases that compose the working activity. This template includes, for each phase, the proper risk calculation method (if any), and to input, for each selected method, those parameters that are not supposed to change among workers and that are inherent the working activity. The segmentation module takes the template from the task analysis and provides a segmentation of the current activity based on a neural network. This module allows the rater to manually segment data to train the neural network. Data transmitted by the devices are received on the host PC by means of data receive software, that can be either custom or commercial ones depending on the device. These data are pre-processed and sent to both the segmentation module and the ergonomic evaluation module. Based on the pre-processed data, the outcome of the segmentation, and the method selection and parameters, provided in the task analysis, the ergonomic evaluation module calculates the risk score for the current phase. This score can be visualized online by means of a GUI. At the end of the capture session(s) the rater can run offline analyses to assess the intra-subject and inter-subjects variability.

### 3.2. Activity Evaluation Procedure

This Section presents the different steps of the activity evaluation procedure. Figure 2 show the flowchart of the procedure developed to support the raters in the computation of the ergonomic risk.

The evaluation procedure begins with the selection of new working activity. The first step of the procedure is the task analysis, that provides the system a template and information for both the segmentation and the ergonomic evaluation modules. The following step is the selection of the wearable devices among the commercial and the custom choices depending on the application fields that can have different critical issues. Then, the data capture for the first subject is implemented and this is followed by manual labeling of the working activity and training of the neural network. The rater performs the labeling of the different work activity phases only for a limited number of repetitions which decreases as the number of subjects increases. In this step, the aid of a technician may be needed to help to adapt the neural network (NN) structure, that is, the input vector and the hidden layer size. Manual labeling stops when the achieved precision is at least 85% on average over the phases and the minimum precision is at least 80%. The next steps are automatic: the system computes the risk scores according to the selected methods and returns it on a graphical user interface. When a new subject is recruited for the evaluation, the flowchart is iterated at the “Data capture” level. The automatic segmentation is performed on the acquired data and checked by the rater. If the precision is not sufficient, data are manually segmented and the NN is trained as in the previous step. The flow is repeated this way for all new subjects. All data (from raw data to ergonomic risk scores) are stored for eventual refinements and offline analyses, that is the final step of the procedure.

### 3.3. Task Analysis

The task analysis is the fundamental step that allows determining the phases and their characteristics. This step enables the definition of the NN devoted to the segmentation and to associate the proper risk assessment method to each phase. In the task analysis, the rater inputs the phases that are included in the target working activity. These phases are used both for the activity segmentation, and for the ergonomic risk evaluation.

For each phase, the rater selects the proper assessment method following the indications of the TR12295 according to the scoring methods available in the system.

The TR12295 is structured in two levels: the first contains some “Key Questions” to guide the rater in selecting and then applying the right ergonomic method depending on the job condition; the second level contains a “Quick Assessment” to help the rater to understand the seriousness of the risks. In particular, after a preliminary screening of the phases, the rater decides whether a risk is possible for such a phase. If so, the prevalent feature of the phase is selected. This means that the rater decides if the main feature of the phase is lifting/carrying a load, push-pull of a load, or a repetitive action at high frequency and with a small load (under 3 kg). In the first case, the NIOSH LI is selected in the task analysis as method to be used for scoring the ergonomic risk. In the second case the Snook & Ciriello method is selected. In the latter case both the REBA and the Strain Index are selected by the rater.

For each of the four aforementioned methods, there are some variables that the rater must provide in the task analysis, as they do not change during the task execution, or because they are difficult to estimate. The other variables are calculated automatically by the system. Section 3.5 reports all the variables needed to obtain the scores in the four cases along with a description that includes whether they are provided in the task analysis or they are calculated by the system.

As a last step of the task analysis, the rater selects the devices that are suitable to perform the capturing activities depending on the conditions of the environment in which the task is executed. Section 3.4 reports the conditions in which each devices can be used.

### 3.4. Devices

The proposed methodology moves from data gathered from a Wearable Sensor Network (WSN), as shown in Figure 1 and Section 3.1. This WSN is composed of an inertial motion capture system and an sEMG system. Different devices are available for both tasks and the selection is based on the working activity and the environmental conditions where data capture must be performed (Section 4.1).

#### 3.4.1. Wearable Inertial Systems

Inertial systems are based on 9-axis IMUs in which the 3D accelerometer, the 3D gyroscope, and the 3D magnetometer measure the linear acceleration, the rotational velocities, and the Earth’s magnetic field, respectively. IMUs include sensor fusion techniques that combine data from the three sensors to provide measures of 3D position/orientation and heading. The use of this technology has different issues whose importance depends on the kind of the algorithm chosen and on the application field. The Xsens MVN suit (Figure 3a) is composed of 17 IMUs placed on the human body segments (head, sternum, shoulder blades, upper arms, lower arms, hands, pelvis, upper legs, lower legs, foots) thanks to elastic bands and connected to a central unit, called Body Pack (BP), that in turn is connected to the battery pack that can last until 9.5 h. The central unit retrieves the data from the different sensors and ensures exactly synchronized samples. It contains a Wi-Fi module that sends the collected data to an external unit thanks to the connection to the proprietary Access Point (AP). This suit is able to send data at 240 Hz and thanks to the proprietary software, the MVN Analyze, the movement of the person wearing the suit are monitored in real-time. Besides, this software allows us to select the calibration procedure modality and to save the data in different formats: the Biovision Hierarchy (BVH), the Coordinate 3D (C3D), or the MVNX. The data capture is configured to obtain 3D position and orientation as well as the angular velocity of all the 23 segments along with the timecode, essential for the synchronization of data coming from other wireless systems such as the EMG device in this study.

For the sake of affordability for the raters, it is possible to use less expensive devices. Although the Xsens MVN is a very robust system, it is also very expensive. Thus, the Perception Neuron (Figure 3b), which guarantees a great compromise between the cost and the technical specification, is considered too. This wearable system is composed of a maximum of 32 IMUs, also called neuron, that are placed on the body thanks to elastic bands and then connected to a central unit that filters and transmits via Wi-Fi the data to a host PC. Unlike the Xsens, this device does not need the use of a proprietary AP, but like it, it needs to be used with its proprietary software, the Axis Neuron, that allows calibrating the system, to monitor the data in real-time and to acquire the data in the BVH format. The data can be transmitted at a maximum rate of 120 Hz using 17 IMUs (the full-body neuron setup without the hands fingers neurons that is enough for an ergonomic assessment of the whole body) attached to the body segments as the Xsens. Unfortunately, the Axis Neuron does not allow saving the timecode together with the inertial data. Thus, a software module have been developed to intercept the data stream of the proprietary software and add the timecode needed for the synchronization with the sEMG data. Both the Xsens MVN and the Perception Neuron systems require an initial calibration procedure that lasts a few minutes.

Finally, the custom device, described in References [32,33], is composed of 11 IMUs, attached to the upper arms, lower arms, upper legs, lower legs, pelvis, head and T1 vertebra, that allows reconstructing the movement of the whole body except the wrist joint. In Figure 3c a setup with four units worn by a volunteer is presented. The data can be transmitted via Wi-Fi to an external unit at 100 Hz and thanks to the server specifically developed the data can be saved together with the timecode. For this device, as presented in Reference [34], a calibration algorithm, an anatomic variables estimation, and a reconstruction of the IMUs motions are developed. The calibration algorithm involves a first step where the user must rest in N-pose (neutral pose in which the person is standing with the arm relaxed to each side of the body and the thumbs point forward) and then joining his/her hands. This procedure allows estimating the sensor pose and the limbs’ lengths thanks to a Kalman Unscented filter. This is the less robust device of the three motion capture systems.

Another difference among the three wearable inertial systems is that the Xsens has only one sensor configuration. It needs all the 17 units to be positioned on each body segments to work. Instead, both the Perception Neuron and the custom device can be configured in various sensor setups if needed. Thus, depending on the application field or on the ergonomic risk scores that needs to be implemented different choices can be done in order to have an efficient, but simpler setup. If a full body configuration is chosen, as in this study, almost every MMH activities can be monitored, that is, port activities, assembly line activities, construction industry activities, nursing activities and so forth. In other application fields, different IMUs setups can be used to monitor only a part of the body. For instance, for bus or train drivers and super market cashier [29], the upper body setup can be used to simplify the overall system and evaluate the ergonomic risk factors of the arms. Refer to Table 1 for an overview of the characteristics of the inertial devices considered.

#### 3.4.2. Wearable sEMG Systems

Regarding the EMG systems, we compared a custom and a commercial device, that is, the Shimmer3 EMG. The first having better performances, but the latter being more robust. The custom device (Figure 4a), an evolution of the device presented in Reference [31], is able to monitor the effort level of the operator’s arms employing eight monopolar sEMG signal channels and an onboard filtering algorithm. Besides, the signals can be monitored in real-time on an external unit thanks to the software specifically developed that wireless (Wi-Fi) receives the EMG data together with the timecode. The system is composed of 2 main units: the central unit, which contains the battery charge circuit and handles the Wi-Fi protocol; and the interface unit that deals with the acquisition of the analog signals with a pre-filtering stage. This stage is composed by two filters: a low-pass filter with a cut-off frequency of 96.7 rad/s to filter out the Electromagnetic Interferences (EMIs); and a sinc filter, implemented inside the Analog to Digital Converter (ADC), to remove the aliasing due to the decimation when the acquisition frequency is reduced. The distinction between the two hardware units allows minimizing electrical interference during signal acquisition. The EMG signal channels are complemented by two other electrodes: one for the Right Leg Driving (RLD) circuit to reduce the common-mode interference, and one for the reference. The device has technical specifications (Table 2) comparable to commercial systems, it has a common-mode rejection ratio (CMRR) of −115 dB, a Signal to Noise Ratio (SNR) between 15 and 16 dB, a resolution of 24 bits, a maximum acquisition frequency of 32 kHz, a gain of 1 to 12, and the small dimensions and weight makes it comfortable and easy to wear. This device is a new version of the one presented in Reference [31] where there was no distinction between the central and the interface unit, the sEMG channels were 32, the wireless communication was via Bluetooth.

The Shimmer3 EMG system (Figure 4b) is composed of up to 16 units and each of them has four sEMG electrodes and one reference electrode to measure two differential sEMG signals that are amplified and properly filtered onboard. The Shimmer3 units include an Electrostatic Discharge (ESD) and Radio Frequency (RF)/EMI filters. The units have a gain of 1 to 12, a maximum data rate of 8 kHz, a resolution of 16 bits, and small dimensions and weight that make them comfortable to wear thanks to specific elastic bands. Also, the units can both save the data acquired to an 8 GB micro SD card and send them via Bluetooth to an external unit (Table 2). Both the Shimmer3 and the custom device make use of disposable Ag/AgCl electrodes (Kendall™ Electrodes ARBO) with a diameter of 24 mm and hypoallergenic solid hydrogel as adhesive. As for the commercial inertial systems, the data can be monitored real-time thanks to the proprietary software, the ConsensysPRO, that allows configuring the parameters (e.g., acquisition frequency and communication protocol) of the units, filtering the data acquired, monitoring real-time the data acquired and synchronizing them with the time machine.

#### 3.4.3. Data Preprocessing

Gathered data are filtered depending on the acquisition device. Commercial devices’ acquisition software already includes filtering steps and so the data are not further filtered.

For what regarding motion data, sensor fusion algorithms embedded in the motion capture devices act themselves as low-pass filters for the links’ poses and the joint angles, that are sufficiently smooth to be used for the subsequent ergonomic analysis. Analogously, the Unscented Kalman Filter (UKF) implemented in the custom motion capture device filters motion data to provide a smooth output. In addition to poses and joint angles, velocities are necessary for the NN input. The custom device and the Xsens provide already the required velocity information. Since the Perception Neuron does not provide velocity data, they are calculated from the posture by time differentiation. The root (see Figure 5) and the joint velocities are computed from two consecutive joint samples thanks to the recorded sample timestamps. Furthermore, a fifth-order low-pass filter (cutoff frequency 20 Hz) is applied to smooth joint velocities. These velocities are used along with the joint angles and the root velocity to calculate all the link linear velocities needed for the segmentation (see Section 3.6).

The EMG signals are filtered by a band-pass filter in both the commercial and the custom device. The filter of the custom device is a band-pass filter between 20 and 250 Hz. This filter is implemented to isolate the relevant frequencies for each EMG channel and thus removing motion artifacts. Besides, a notch filter is implemented with 50 Hz to remove electrical interference from signals.

Acquisition and filtering of raw data, as well as their physiological generation, constitute pipelines whose timings must be taken into account to merge kinematic and kinetic data both in the segmentation and ergonomic evaluation. For each device, as shown in Figure 6, it is estimated a times acquisition pipeline that goes from muscular contraction to the machine time assigned to data when elaborated by the NN and the ergonomic evaluation modules. The pipeline considers the latency in signal generation, the conversion and filtering of the data, the communication within the device, and data transmission to host PC. For the host PC, it is considered a common Windows thread execution in 8–16 ms. Thus, in the worst possible case, this time span determines a 16 ms latency that leads to an entire thread execution skip. In the figure, there are reported the three software products (Axis Neuron, MVN, and ConsensysPRO) relative to the commercial devices and the server latency during acquisition data. The wireless latency is estimated in 6–8 ms, this value is calculated considering the round trip time and the number of bytes transmitted. The commercial hardware latencies are reported in the product’s technical specifications. The muscular contraction precedes the actual motion movement recorded by the motion capture systems of a 20–40 ms time span. This latency, known as electromechanical delay, is physiological and it has been deeply studied in the literature (see Reference [35] for an example). It depends on several factors and the assumed values are common for the investigated muscular units activation. This activation is included in the overall latency computation. The data packets in the EMG custom device are generated and filtered by an ADC converter in less than 1 ms and sent with Serial Peripheral Interface (SPI), an interface bus used to transfer data from the ADC converter to CPU. The Universal Asynchronous Receiver Transmitter (UART) provides asynchronous communication from the CPU to a serial Wi-Fi module transceiver. Regarding the IMU custom device, the latency between the signals generation and their availability in the direct access memory (DMA) module in the InvenSense MPU-9250 is overestimated to 1 ms. This latency includes the sampling of the MEMS signal by the internal ADC and their writing in the DMA. The data are sent through SPI to CPU and through UART to a Wi-Fi module transceiver. For both custom devices, the SPI and the UART time latencies are not significant for the final calculation. The highest overall latency is 100 ms, whereas the worst misalignment between motion and EMG data is 83 ms when the custom device and the XSens suit are used together. Since each phase lasts at least one second, the misalignment does not affect the analysis. The latency is deemed sufficient to provide the worker and the rater with usable online feedback. The filtered and synchronized data outputs are then available for the segmentation and ergonomic evaluation modules.

### 3.5. Calculation of Variables for ISO11228 Methods Implementation

Full-body motion capture and a correct data analysis allow the acquisition of all necessary information to satisfy standard assessment methods listed in the ISO 11228 standard, that inclue the NIOSH lifting index, the Snook & Ciriello method for pulling and pushing, and the REBA and the Strain Index for repetitive actions (see Section 2.1). Although several formats are available for data communication and storage, in this study we consider to parse the BVH format independently of the device. In particular, a BVH file is composed of two parts: a header section that describes the hierarchy and the initial pose of the skeleton and a data section that contains the motion data. The skeleton is represented through a hierarchy, the parent-children relationship between links that originates from a reference or root node. Each link is connected to the son through a ball joint. In the same section, there are also stored the spacial reference variables for each link. The workers’ anthropometric variables are set earlier to the start motion capture session. Furthermore, the data section includes the number of frames, the mean frame rate, and the motion data matrix. The matrix columns are ordered according to the skeleton hierarchical links sequence, while the rows follow the temporal sequence for each variable, reporting the motion variables. The reference six parameters are needed to identify position and orientation. For what concerns the other nodes, they are described using three Euler angles measurements. These variables provide the pose, position, and orientation, of each link according to the world reference. Figure 5 shows the kinematic chain used for motion reconstruction. Connections among links are modeled as spherical joints.

#### 3.5.1. Lifting Index of NIOSH

The implementation of the NIOSH method allows us to analyze lifting and carrying movements. The NIOSH method requires the calculation of several factors (see Table 3) that depends on the user’s posture and the frequency of actions. The method requires the computation of the Lifting Index, defined as:(1)$$LI=m_{A}/m_{r},$$
where $m_{A}$ represents actual lifted mass and $m_{r}$ recommended limit mass. If LI>1 determines an unrecommended condition. $m_{r}$ is obtained via the product:(2)$$m_{r}=m_{ref}\cdot h_{M}\cdot v_{M}\cdot d_{M}\cdot \alpha_{M}\cdot f_{M}\cdot c_{M},$$
$m_{ref}$ is the reference mass for the identified user population group; $h_{M}$ is the horizontal distance multiplier; $v_{M}$ is the vertical location multiplier; $d_{M}$ is the vertical displacement multiplier; $\alpha_{M}$ is the asymmetry multiplier; $f_{M}$ is the frequency multiplier; $c_{M}$ is the coupling multiplier for the quality of object gripping. The reference mass $m_{ref}$ is set based on workers age and gender. The horizontal distance is conceived as the mean of the horizontal distance between the midpoint of WrR and WrL (see Figure 5) and the midpoint of ToeR and ToeL during the lifting task. The minimum hands-ground distance measurements are considered as the vertical location of the object lifted. The vertical displacement is estimated through hands movement considering the difference between the minimum and maximum height of the WrR and WrL midpoint. The asymmetry angle or twisting requirement is estimated through the relative orientation of the chest link with respect to the root link. Frequency and duration of lifting activity are estimated from timestamp data during lifting actions. The evaluation of the quality of the grip is provided by the rater during the task analysis. Also, these parameters need to be converted according to specific conversion tables, provided with the NIOSH method, in order to calculate the proper multiplier factors which are to be used in the Formula (Equation 2). The actual lifted mass $m_{A}$ is finally provided during the task analysis.

#### 3.5.2. Snook & Ciriello

The Snook & Ciriello method allows analyzing pushing and pulling activities. The Snook & Ciriello tables provide a benchmark for the initial and the sustainable pushing and pulling forces. Three parameters are needed to employ Snook & Ciriello: handle height, covered distance, pushing or pulling frequency, and target worker information (see Table 4). The handle height is evaluated through the height of the WrR and WrL midpoint during movement (Figure 5). The covered distance is calculated in the relevant phase time interval using the position of the Root. Furthermore, the data recorded and the timestamp acquisition enable determining the frequency of the actions. If the ratio between the displaced weight and the recommended weight (based on the workers gender) is over the limit of 1, the risk is defined as unacceptable.

#### 3.5.3. Rapid Entire Body Assessment

The REBA method is divided into 13 steps (see Table 5) that analyze the whole body posture. For an accurate application of this standard, the upper and lower body’s anatomical joint angles must be detected. However, the rotation angles provided by the motion capture devices do not always fit with the anatomical variables needed to fill the REBA checklist. Therefore, these angles are used to obtain rotation matrices between links, and new parametrizations are adopted to obtain the desired anatomical variables. With the first three steps, the rater assigns three different scores based on the head flexion/extension, lateral bending and right/left rotation, the trunk flexion and extension, and knee flexion/extension. The head flexion/extension, the lateral bending, the right/left rotation, and the knee flexion/extension angles are computed evaluating the orientation of the respective links, relatively to their father links. The flexion/extension of the trunk is estimated as the vertical axis related angle stemming from the triangle built thanks to the coordinates of three points. One is obtained through the projection of the Che position on the horizontal axis passing through the Root, the second one is given by the position of the Che itself and the third one is earned knowing the location of the Root (Figure 5). In step 4, the aforementioned anatomical angles are the input of a first intermediate table (Table A) that returns a score $S_{ta}$. In steps 5 and 6, depending on the carried load class, which is provided as input in the task analysis, $S_{ta}$ is increased of a quantity that ranges between 0 to 2 to obtain the “Score A” $S_{A}$. Steps 7, 8, and 9 determine three scores for the shoulder, lower arm, and wrist posture. These scores are based on the shoulder abduction/adduction angle, flexion/extension and internal rotation angles, the elbow flexion/extension and supination/pronation angles, the wrist flexion/extension, and ulnar/radial deviation angles. These anatomical angles are computed evaluating the orientation of the respective links relative to their father links. The 3 scores are used in step 10 as input in Table B, to which a 0–3 Coupling Score must be summed in step 11 as a function of grip’s quality, thus achieving “Score B” $S_{B}$ at step 12. The coupling score is currently provided in the task analysis. $S_{A}$ and $S_{B}$ are the input of Table C and therefore “Score C” $S_{C}$ is achieved. Finally, an Activity Score (based on action frequency estimated from timestamped data) is determined in step 13 to finally get the REBA Score. The REBA score is subdivided into different risk levels varying from a negligible risk with a score of 1 to high risk over 8. REBA Score is evaluated as the worst mean of the scores obtained in the whole time interval between right and left body parts.

#### 3.5.4. Strain Index

For a deeper estimation of the risk for upper limbs, EMG signals are used to implement the Strain Index method and to help to label the worker activities.

The Strain Index method is composed of several multiplier (see Table 6) that include quantitative and qualitative factors associated with verbal cues that include the intensity of exertion $I_{e}$, duration of exertion $D_{e}$, exertions per minute $f_{e}$, posture $P_{m}$, speed of work $S_{m}$, and the duration per day $D_{m}$. The Strain Index score is then defined as
(3)$$SI=I_{e}\;D_{e}\;f_{e}\;P_{m}\;S_{m}\;D_{m},$$
whereas the duration of exertion, the number of exertions per minute and the duration of the task per day are quantitative, the intensity of exertion, the hand/wrist posture and the speed of work are qualitative measures. As reported in Reference [29], some Strain Index factors are calculated directly from the capture system to translate qualitative factors into quantitative measurements. The exertion intensity percentage is computed from exerted effort and Maximum Voluntary Contraction (MVC).
(4)$$I_{e}=\frac{E}{MVC},$$
with
(5)$$E=\sum_{j=1}^{ch}\sqrt{\left(\frac{1}{N}\right)\sum_{i=1}^{N}{\left(x_{j}^{i}\right)}^{2}},\;ch=\mathrm{number}\;\mathrm{of}\;\mathrm{channels},$$
where $\left(x_{j}^{i}\right)$ represents the *i*-th value of channel *j*, and *N* represents the window length. The percentage of the effort intensity permits to classify the effort class according to the converting table of the Strain Index method. The worker’s MVC is obtained from a maximal effort static contraction of the considered muscle group. The measurement of the exertion duration is described as the ratio between the exertion final ($t_{f}^{e}$) and initial time ($t_{f}^{i}$) difference and the cycle final ($t_{c}^{e}$) and initial time ($t_{c}^{e}$) difference, including recovery time.
(6)$$D_{e}=\frac{t_{f}^{e}-t_{i}^{e}}{t_{f}^{c}-t_{i}^{c}}.$$

The number of exertions are estimated from timestamped data during workers activities. The hand/wrist posture is assessed by converting the wrist score presents in the REBA method, with the addition of the Coupling Score factor, according to the following parametrization—the wrist REBA score of 1–2 corresponds to good hand/wrist posture of the SI, the score 3–4 to a poor posture and the score 6 to a very bad posture. Finally, the speed of work is a qualitative measure set in task analysis.

### 3.6. Segmentation and Labelling

The segmentation of the recorded time series is done in two steps. The first segmentation is carried out by the rater (or who runs the data capture session) and it targets the identification of the time intervals in which the effective working activity, that is, one of the phases defined in the task analysis, occurs. The rater marks the initial and the final frames of such intervals by means of a GUI interface. This segmentation allows subdividing between activity and inactivity states. From these segmented and recorded data, a Multi-Layer Perceptron (MLP) performs automatic labeling of every working phase previously identified in the task analysis. This architecture was deemed sufficient after the initial segmentation based on laboratory experiments and an analysis of the literature. As seen in Reference [36], an MLP implemented with 1 hidden layer, 20 neurons, 8 output classes and a sigmoid transfer function accomplishes a 95.6±2.2 accuracy per subject. The data set consists of 15 healthy subjects and 8 different human activities. In this study, the complexity of the activities is similar to the targeted MMH activities. Therefore, the methods used for the activity segmentation as well as the adopted parameters have been used as a reference for the construction of the proposed NN and to estimate the dataset size needed for its training. A preliminary investigation carried out in the laboratory confirmed that the structure is suitable to accomplish the goal. This activity identified the most relevant skeleton links apt to improve the neural network abilities to recognize MMH activities. In addition, an increase in the neural network accuracy was observed especially when including the EMG signals in the input vector to classify workers’ actions. Therefore, the neural network input is a vector subdivided into motion and EMG data sets. Data are resampled to accomplish the same sample rate among different devices. Motion data typically include the positions and the velocities, relative to the chest, of the ankles, the hands and the pelvis along with selected joint angles and velocities when needed.

For each EMG channel, the signal’s Root Mean Square (RMS) is taken into consideration to composing the input vector together with the motion data. The size and composition of the input vector as well as the size of the MLP vary depending on the working activity. Section 4 reports a practical implementation for a representative case study. The MLP neural network is set up by means of both the task analysis and part of the recorded data from each new worker. The setup activities include limited adjustments to the input vector and to the number of hidden layer neurons. The neural network accuracy depends on the number of different workers considered and the working session duration for each worker. The greater the number of output classes to be recognized by the network, the more data is have to be collected to set up and train the network. The NN performance improves as the number of available workers thanks to increased data variability. Future work will investigate the opportunity to replace the MLP with autoencoders to avoid the need for online segmentation of working activity time intervals.

### 3.7. Analysis and Visualization of the Results

#### 3.7.1. Online

Four graphical interfaces have been developed for each considered assessment method, that is, NIOSH, Snook & Ciriello, REBA, and Strain Index. This kind of information helps the rater to monitor the risk score and the worker’s main anatomical variables during the working activity and potentially give feedback to correct the worker’s behavior. All these interfaces, except the Strain Index interface, show online the workers’ posture and movements, and the main scores associated with the anatomical variables. Figure 7 depicts the NIOSH graphical interface as an example of online output.

#### 3.7.2. Offline

Other important elements are provided to expert operators with an offline data analysis. First, the frequency of the workers’ activities is extracted from timestamp data of each phase making its distribution over the capturing session available to the rater. Moreover, the average and the variability of the risk scores and of target variables (e.g., back flexion/extension) can be extracted by the rater to make intra-subject and inter-subjects analyses. The analysis of variability is fundamental to assess whether the biomechanical overload risk is inherent to the task or it is rather due the particular characteristics and behavior of a worker. Moreover, the intra-subject variability allows to assess whether a worker’s performance is constant in time or it it changes. Changes may be due varying conditions of the task (e.g., progressive lack of room in logistics applications) or fatiguing of the worker. This latter factor may be also investigated by means of the available data: both kinematic and EMG data allow indeed for investigating the presence of fatigue in the task execution. The raters have also direct access to the intermediate multiplier factors and scores of the implemented methods to identify the main contributions to the ergonomic risk. These steps allow the rater to investigate numerically which factors are mostly affecting the risk score, providing the rater useful insights to further investigate the task, even for a specific worker, and devise effective changes.

### 3.8. Estimation of the Analysis Time for the Proposed Method and the Traditional One

The time needed to perform a complete evaluation of the working activity of one worker has a large impact on the number of workers that can be considered in the evaluation of the activity. Therefore, for the assessment of the proposed method, the total evaluation time will be considered in addition to the correctness of the risk score. In particular, this section proposes a general, approximate comparison of the time needed to complete the evaluation of *n* workers with a traditional, purely observational method and the proposed one.

The traditional method moves from the task analysis and it is composed of three steps: data capture, segmentation, and checklist filling, thus obtaining the ergonomic risk. The proposed method moves from the task analysis as well, and it is composed of three steps: data capture, segmentation, and calculation of the ergonomic risk.

The task analysis takes the same amount of time for the two methods, it can be estimated in half working day (4 h), and it is needed once for each new activity. Data capture usually lasts 2–4 h of working activity recording. Only a part $t_{r}$ of this time, typically 15 to 25%, is spent in the relevant phases for the ergonomic risk. In both the traditional and the proposed method, data capture includes a briefing session to inform the worker about the recording and its scope, which lasts 5 to 10 min. In the case of the proposed method, an additional 20 min must be taken into account for the worker to dress the devices, calibrate them, and for undressing at the end of the capturing session.

In the traditional approach, data segmentation consists of manual labeling of the whole-session time series independently of the number of subjects. Instead, in the proposed method the segmentation is composed of three sub-steps: manual labeling of part of the time series, training of the neural network, and phases classification. In both cases, manual labeling takes about three times of the analyzed time interval $t_{a}$. In the traditional approach $t_{a}=t_{r}$, whereas in the proposed method $t_{a}=k_{i}t_{r}$, where $k_{i}<1$ and it decreases as more workers are analyzed. A preliminary analysis on MMH activities composed of 3–6 phases led to estimate that $0\leq k_{i}<0.8$, and that $k_{i}=0$ for i>8. Training of the neural network takes approximately 15 min for each new worker using the material and the approach described in Section 3.6. Once the network is trained, segmentation takes from 1 to 3 milliseconds per minute of analyzed data. In the traditional approach, the last step requires the rater to observe the videos of each analyzed phase and manually annotate all the variables, angles, displacements and times, required as input to the risk assessment algorithms. This takes the rater a large amount of time that is estimated in 5 times $t_{a}$. This estimation accounts for the expertise of the rater, who can select significant executions of the task to produce the assessment. If the rater would annotate the whole segmented data, this time would be roughly 20 times $t_{a}$. The proposed method does it automatically based on the input from gathered data, segmentation, and task analysis. This takes less than 1 millisecond per analyzed frame.

The total time needed for the three steps of the two procedures is reported in Figure 8 along with an estimation of the total time needed to process up to 10 workers for the same activity. A conservative assumption of two hours of capture per worker has been made. Despite the initial time cost caused by the NN training, as the number of workers increases, the segmentation time consumption decreases notably, leading to a time advantage in the overall segmentation phase, that adds to the one of the ergonomic risk calculation. Therefore, the proposed method saves time even for a few workers, and its advantages grow considerably for a larger number of analyzed workers.

## 4. Case Study

For the evaluation of the proposed system, this section reports the application of the proposed method to a challenging port activity such as Lift-on/Lift-off of containers. In particular, it deals with the Lift-on/Lift-off (LoLo) ships that are cargo ships loaded and unloaded with cranes. Then once the container is on the cargo ship, the operators handle the lashing/unlashing operation phases by means of specific tools.

As reported in Section 3.3, the task analysis of the fasten/unfasten procedures is performed by selecting the phases of this activity that present ergonomic risks for the workers in terms of biomechanical overload. Thus, the suitable ergonomic evaluation method for each phase is selected. The procedures include both the lashing and the unlashing operations when the cargo ship is loaded and empty with the containers respectively. Anyway, these opposite phases are similar since the operators’ movements are almost the same. The LoLo procedure (Figure 9) consists of three phases:**Phase 1** Lifting/Lowering the rod and hooking it to the container**Phase 2** Lifting/Lowering the swivel**Phase 3** Tightening/Untightening the swivel to the rod

The Phase 1 is performed four times for each container, instead the others twice for each. In addition, more operators are working simultaneously on the cargo ship and they often work together on the same container so that the frequencies of each action, needed for the implementation of the ergonomic indices (Section 3.5), have a high variability.

### 4.1. Experiments

Given the difficult environment, the commercial devices presented in Section 3.4 have been preferred. The main issue to be considered in port activities is the interference due to the magnetic fields that inside a cargo ship are considerably high since the workers are completely surrounded by metallic objects. Thus, a preliminary test inside the cargo ship is performed by acquiring the motion data with both the Xsens MVN and the Perception Neuron and, as a result, the Xsens MVN has been chosen because it showed to have a higher resistance to magnetic field disturbances than the Perception Neuron (Table 1). In addition, in this application fields the full-body IMUs sensor setup is needed since port workers can incur in various biomechanical overload risks. As reported in Section 3.4.1, the 17 Xsens IMUs are placed on all the body segments.

The data acquisition sessions lasted around 2 h and involved 2 volunteers, one for the lashing phase and one for the unlashing phase. As a first step, the worker was informed about the research activity in which the experiment is performed, and the usage and dissemination modalities of the acquired data. Thus, before starting the acquisition session, each volunteer has given his consent signing the informed-consent document approved by the ethics committee and previously explained to him. Then, the volunteer was dressed with the wearable devices presented in Section 3.4: the Xsens for the acquisition of the whole-body movements and the Shimmer3 for the forearm EMG signals acquisition. This phase lasts around 10–15 min.

Two Shimmer3 units were employed and they were fastened to the right and left wrists of the volunteer with two elastic bands. On both arms, the reference electrode was placed on the medial or lateral epicondyle by occurrence and the differential electrodes were placed on the superficial finger flexors as depicted in Figure 4b. As seen in Section 3.4, the Shimmer3 units are provided with Bluetooth communication that is reliable within a maximum distance of 6–10 m, depending on the obstacle between the units and the external device. Thus, it is important that the distance between the volunteer and the external device acquiring the data is not higher than the aforementioned distance. Instead, for the Xsens MVN suit, that transmits the data to the external device via Wi-Fi, the distance restriction is less tight and broadly sufficient for the planned activities. The volunteer’s anthropometric variables are measured and the three steps calibration procedure, that is needed to use the Xsens MVN, is described to the worker and then performed. Figure 10 show the sensors worn by the user.

The calibration procedure is run as follows. In the first step, the volunteer must stay in the N-pose. In the second step, the volunteer must walk back and forth following the MVN Analyze software guide explained by the experimenter. Then, the volunteer must go back in the first position maintaining the N-pose while the experimenter accepts the calibration. This phase lasts around 15–20 min. Then, after checking that the external devices are receiving the data properly, the volunteer performs the work activities as usual.

For the wireless data acquisition, in addition to the wearable devices, the two experimenters used the following equipment: a tablet DELL Venue 11 Pro with ConsensysPRO software installed to acquire the sEMG data via Bluetooth; a laptop DELL XPS 15 with MVN Analyze software installed to acquire the inertial data via Wi-Fi; a UPS Trust as a portable power supply; and a router Asus AC3100 to make the Wi-Fi network for the inertial data transmission. In addition, before starting the acquiring session, the tools used by the workers for the lashing/unlashing operations are weighted and their grasps are evaluated. During the acquiring session, the experimenters check that the wearable devices remain in the right position and that the data are received properly both on the tablet and on the laptop.

The proposed MLP is based on the approach reported in Section 3.6 and it has been adapted to the Lo-Lo activity. Several NNs have been trained with a number of nodes between 8–20. The Neural Network outputs relocate input information to the class to which it corresponds. The input vector length is composed of 962 elements. 750 elements are extracted from motion data. In particular, they correspond to 50 samples of the velocities and positions, both relative to the chest, of the ankles, hands, and pelvis. The remaining 212 elements include 53 samples of the RMS of the EMG signals gathered from the 4 electrodes placed on the forearms, using a moving window of 20 samples. In its totality, the vector length is equivalent to a 100 ms duration time. The NN is trained by means of a dataset split randomly at each iteration: 70% of the sequences are used to train the model and 30% for the validation test. The recorded data has an overall duration of 2 h (60 min for each subject), of which 18 min are obtained from segmentation activities and need to be labeled by the NN. Namely, in these 18 min, 16 min of recorded activity is considered to train the NN and 2 min are taken for the test. The dataset for the training and validation set includes 103 rod liftings/lowerings (phase 1), 45 swivel liftings/lowerings (phase 2) and 50 swivel to the rod tightenings/untightenings (phase 3). The test set includes 10 rods liftings, 7 swivels tightenings and 7 swivel liftings.

### 4.2. Traditional Evaluation

The same activities are shown in Section 4 have been evaluated by three human raters from video captures with the purely observational approach. The raters segmented the videos annotating the time intervals of the relevant phases and the variables required to obtain the NIOSH LI and the REBA score in Excel files. In these files, the rater fills in manually several predisposed cells with values corresponding to the parameters requested by the evaluation methods.

### 4.3. Results

Three human raters performed the traditional evaluation expressed in Section 4.2, on a 2 h video, 1 h for each subject. The raters performed the assessment on the whole session, but they provided also a feedback after the analysis of selected executions of the working activity. The average duration of the whole assessment is 9 h 30 min for the three raters. However, they were ready to perform the assessment after 3 h on average. This time is subdivided into 2 h of data capture, 62 min for labelling and the remaining time for checklist filling. Conversely, in the proposed method, the total time needed is 4 h 18 min and it is composed as follows. The segmentation activity is performed during the working session for an overall time of 2 h, 25 min per subject were needed to administer informed form, dress of the devices, instrument calibration and undress. The manual labeling took 54 min to obtain 16 min of labeled captured data in which the relevant phases occurred. Further 2 min of activity were labelled by the trained NN without manual labeling. Finally, 14 min is the time that was needed to train the aforementioned variants of the proposed MLP neural networks and select the optimal one. The selected NN provides the best trade-off between the accuracy and the risk of overfitting of the training dataset. The confusion matrix reported in Figure 11 shows the performance of the selected MLP with one hidden layer composed of 13 nodes.

The offline data processing favours the intra and inter subjects analysis showing the mean value and the variability during the entire working sessions. The manual evaluation has obtained the following NIOSH LI and REBA scores: for lashing, 1.96±0.41 in phase 1, 0.56±0.06 in phase 2 and 4.13±1.40 in phase 3; for unlashing 1.46±0.16 in phase 1 and 0.49±0.08 in phase 2 and 4.66±0.94 in phase 3. These results are reported along with the proposed method scores to ease their comparison. Figure 12 shows the NIOSH score obtained through either automatic or traditional evaluation during workers lashing and unlashing activities.

In addition to the aggregate scores, the proposed system allows to highlight the factors that determine the result. Figure 13 shows the multiplier factors and their variability. Similarly to the NIOSH LI, the systems allows the rater to analyze more in depth the outcome of the REBA score. Figure 14 shows the scores of the REBA method related to the postural factors, that is, steps 1, 2, 3, 7, 8, 9, related to neck, trunk, legs, upper arms, lower arms, and wrists respectively.

The scores calculated by means of the REBA and the Strain Index methods to evaluate the phase 3 are presented in Figure 15 and Figure 16 respectively.

### 4.4. Discussion

The results of the experimental activities show the effectiveness of the system for what regarding both the coherence of the scores and the overall time needed to perform the evaluation. The scores reported in Figure 12 and Figure 15 are consistent with the traditional method outcome in all cases. The same figures report two kinds of variability—the variability of traditional rating and the variability of the proposed method.

The first variability is intrinsic in the qualitative visual assessment of postural variables that can nowadays be measured consistently. This variability can be reduced by increasing the expertise of the rater or improving the quality of the video recording. However, it cannot be eliminated. The variability of the proposed score depends only on the execution of the task performed by the subjects. It can be therefore used to assess both the intra-subject and inter-subjects variability. The first can be used to tailor corrective actions to the specific worker. The second can be used to assess whether the ergonomic risk is intrinsic in the activity, or the physiological and behavioral characteristics of individual workers impact the risk significantly. In this latter case, the lowest risk conditions can be used to devise interventions for the execution of the task, or bad combinations of physiological and behavioral characteristics can be identified to train workers that are subjected to a higher risk or redirect them to other tasks.

The advantage of the proposed method with respect to the traditional one is apparent from the analysis of Figure 13 and Figure 14. Thanks to the high reliability of the measures obtained from the wearable devices, the value of the factors that determine the risk score and their variability can be used to deepen the analysis of the risk associated with the working activity. For example, Figure 13 shows that in both phases of lashing and unlashing the horizontal displacement of the carried object plays a very important role, even if the task is mainly a lifting task. The variability of the factor shows that the initial (in case of lashing) and final (in case of unlashing) position of the rod on the ship vary, and it depends only on the task execution. Therefore, a specific intervention on how rods are left on the ship may reduce this factor and the biomechanical overload risk. Similarly, Figure 14 shows that the lower arms and wrists are the most contributing to the overall risk, as it could be expected. However, it is interesting to note that the posture of the upper body (trunk and neck) vary significantly among workers and task executions. This information could be used to devise a method for the execution of the task that minimizes the ergonomic risk. The Strain Index reported in Figure 16 is consistent with the REBA score and it reports a higher risk as expected: the role of the upper limbs is indeed amplified by the Strain Index with respect to the REBA score.

The developed neural network showed to be able to correctly classify the phases of the analyzed task. The precision meets the proposed target. However, the obtained results are insufficient to evaluate the effort that would be necessary to extend the analysis to further workers on the same task, and to transfer the learned classification capabilities to other activities. Anyway, the time needed for segmentation without annotation is still significantly shorter than the time needed to both segment and annotate data. This make the development of an efficient segmentation tool in terms of setup time, expertise required for the setup, and precision, important but not central for this system. Therefore, further investigation on the segmentation block have been postponed with respect to the development of the proposed method.

The comparison of the measured times is consistent with the forecast reported in Section 4.2 and it confirms that the proposed method requires less time to be executed with respect to the traditional one, thus allowing the rater to analyze more subjects and to focus on the analysis of the results rather than on the annotation of the videos. In the results, this happens already for the analysis of just one worker. In fact, it took a small amount of time to adapt the proposed network to this activity, as shown in Figure 11. However, the break-even point between the traditional method and the proposed one may occur later if the set up of the neural network would take more time. This is the main weakness of the proposed method. In fact, if the MMH activity is particularly complex, the set up of the neural network may require time and the intervention of an expert. This is the reason why further research activities are ongoing to devise a robust and easy to tune algorithm for automatic segmentation of the MMH activities. However, time advantages still occur if the rater segments the activity manually, while keep the huge advantage to have reliable, repeatable, and objective measures of both postural and muscular activation variables.

## 5. Conclusions

The paper presented a novel system for the evaluation of the ergonomic risk due to biomechanical overload. The method advances the state of the art as it includes the EMG for the evaluation of the effort and it is extended to the whole body. The system has shown its potential in terms of identifying a specific behavior of a worker and its variability, and the factors that influence most the ergonomic risk. The experimental results show that the system can be used on complex scenarios such as a cargo ships obtaining results consistent with the traditional risk assessment methods. The measures obtained allows the rater to deepen significantly the risk analysis by considering both the intra-subject and inter-subject evaluation. Moreover, the combination of motion and EMG data can be used for further specific biomechanical analyses. Thanks to the developed interfaces, the system can be easily adopted by raters without specific expertise in programming. The data capture procedure is also easy to learn by means of a few practice sessions. The main drawback to the use of the proposed system lies in the segmentation of the activity, that may require the intervention of an expert. However, time advantages still occur even if the activity is segmented manually, in addition it leads to objective measures of the motion and effort variables.

## Figures and Tables

**Figure 1 sensors-20-03877-f001:**
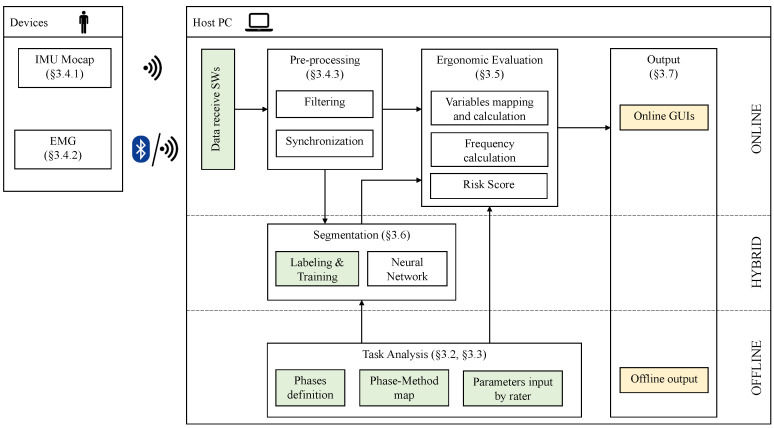
System Architecture. On the left side wearable devices and the communication with a host PC. On the right side the modules that compose the system, from the task analysis to the online segmentation and processing of kinematic and EMG data to obtain the risk scores. Green blocks represent software modules where the input of the rater is required. Yellow blocks represent modules that provide scores and output of the analyses available to the rater.

**Figure 2 sensors-20-03877-f002:**
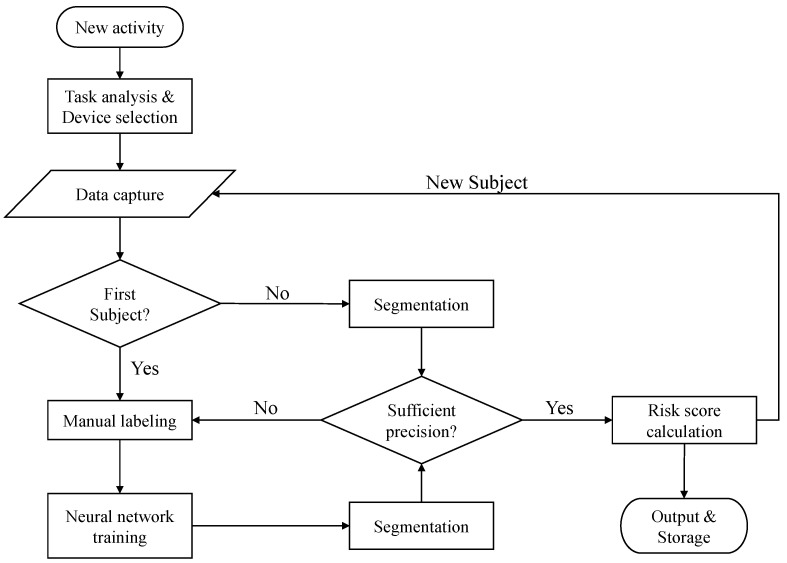
Flowchart of the risk evaluation procedure for a new working activity. The procedure takes into account possible iterations on the training of the neural network for automatic segmentation of the activity.

**Figure 3 sensors-20-03877-f003:**
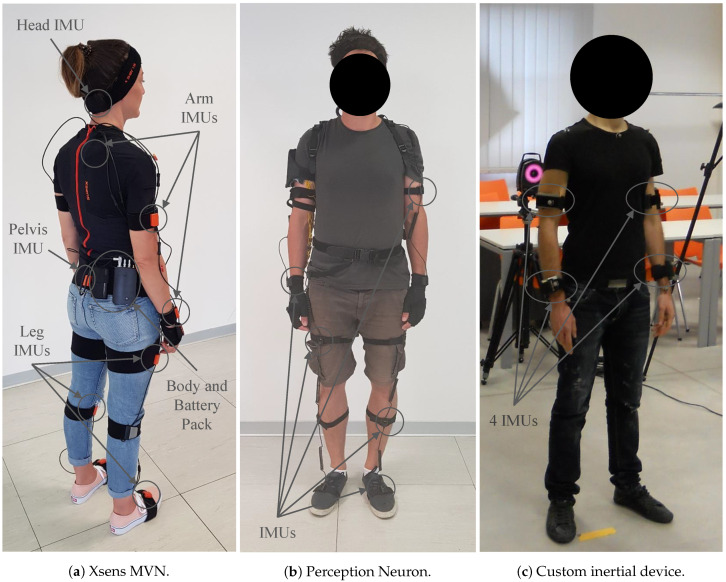
Wearable inertial devices worn by volunteers.

**Figure 4 sensors-20-03877-f004:**
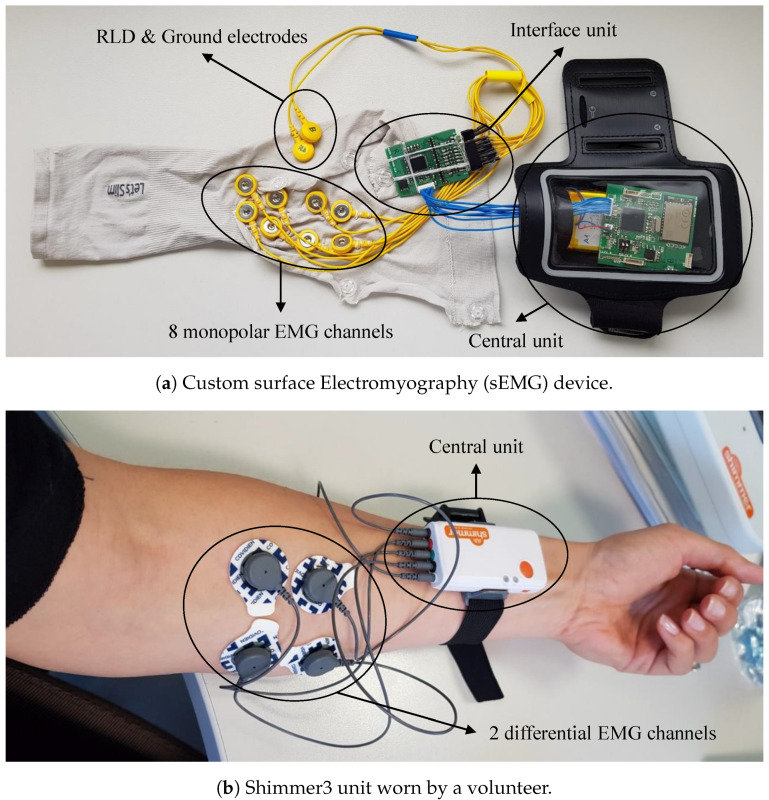
sEMG devices.

**Figure 5 sensors-20-03877-f005:**
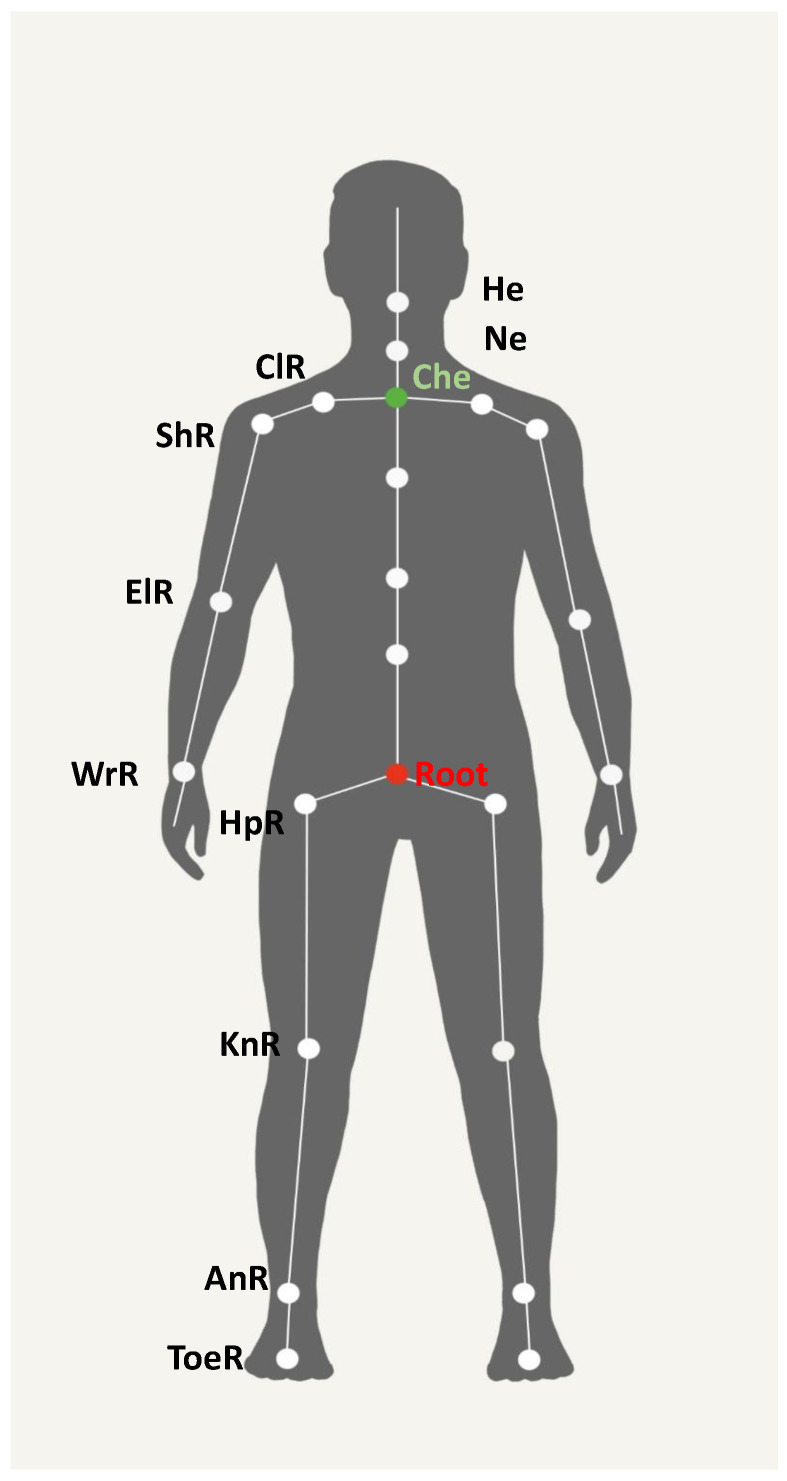
Kinematic chain used for motion reconstruction. Centres of the spherical joints coincide with the origins of the frame attached to the child link following a hierarchical structure rooted in the waist. The names of these origins for some relevant frames are reported. Only the right side of the human body has been reported for symmetric links. The listed joints refer to: root (Root), chest (Che), neck (Ne), head (He), collar (Cl), shoulder (Sh), elbow (El), wrist (Wr), hip (Hp), knee (Kn), ankle (An) and toe (Toe)

**Figure 6 sensors-20-03877-f006:**
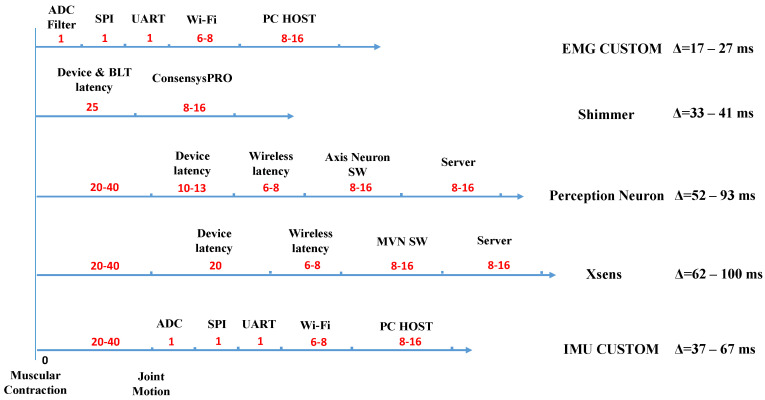
Comparison between the devices latencies. Listed below there are the acronyms used: Analog to Digital Converter (ADC), Serial Peripheral Interface (SPI), Universal Asynchronous Receiver Transmitter (UART), Bluetooth (BLT) and Moven Software (MVN SW).

**Figure 7 sensors-20-03877-f007:**
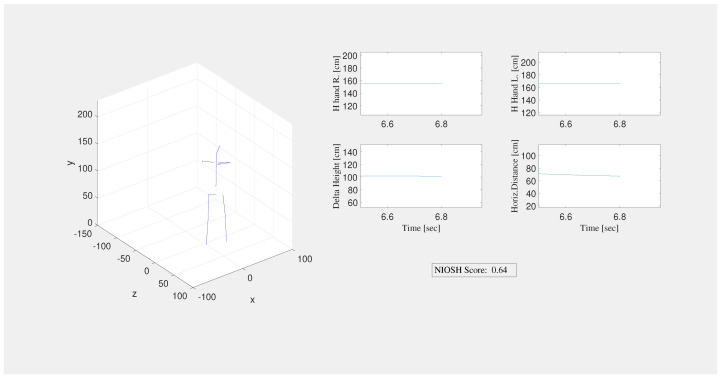
NIOSH Interface.

**Figure 8 sensors-20-03877-f008:**
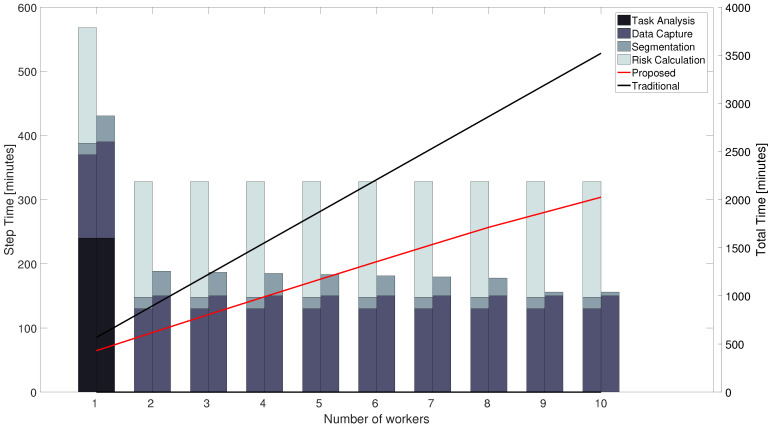
Foreseen temporal structure and duration comparison of risk evaluation for the Traditional and new method. Column groups report traditional evaluation on the left column and the proposed method on the right column. The assumption that an expert rater does not need to annotate the whole recorded activity to score the risk has been adopted.

**Figure 9 sensors-20-03877-f009:**
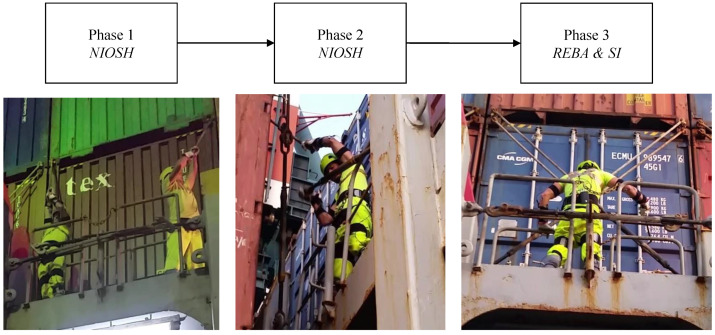
Lift-on/Lift-off (LoLo) phases and ergonomic methods selected.

**Figure 10 sensors-20-03877-f010:**
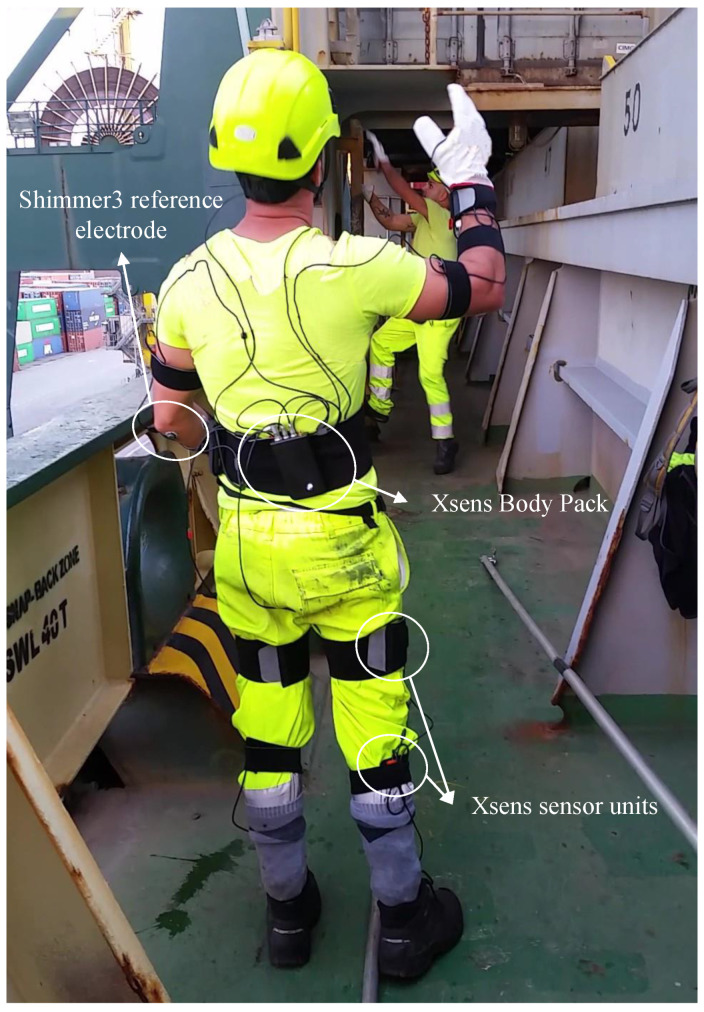
A volunteer ready to start his work shift wearing the wearable devices. The sensor units have been placed on the body segments as described in Section 3.4.1.

**Figure 11 sensors-20-03877-f011:**
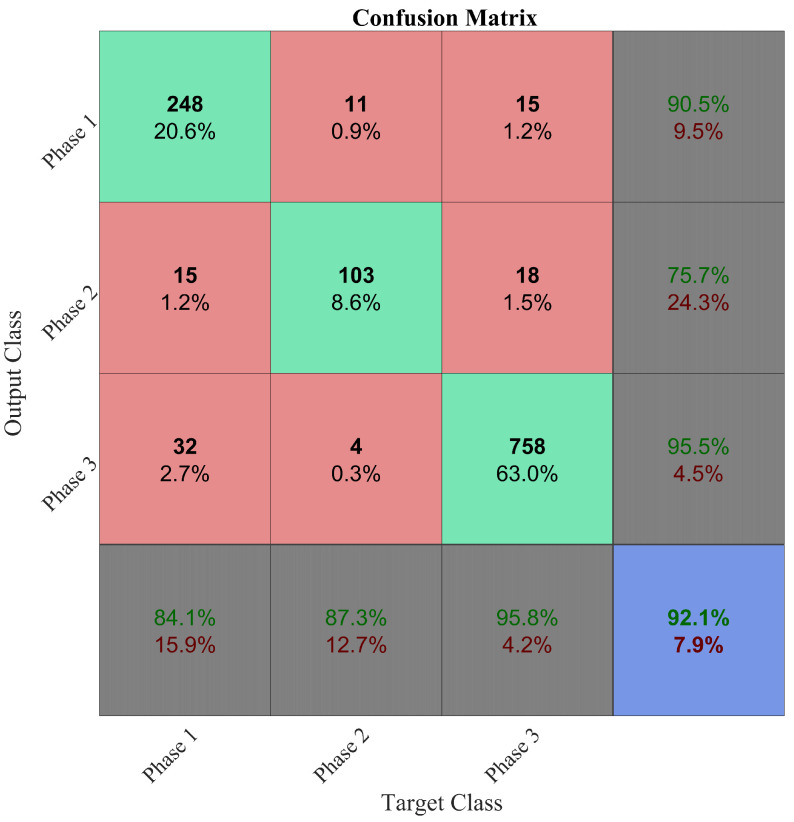
Multi-Layer Perceptron (MLP) Neural Network performance. The rows refer to the predicted class (Output Class) and the columns correspond to the true class (Target Class). The last column represents the percentages of all inputs correctly and incorrectly predicted for each class in green and red respectively. The last row represents the percentages relative to the target class that are correctly and incorrectly classified in green and red respectively. In the inner 3×3 square, the diagonal cells green squares correspond to correctly classified inputs, the other cells (red squares), to incorrecly classified inputs. The black numbers and percentages refer to the number of elements for each class and their percentage over the total dataset for the network test. The overall accuracy is reported in the lowest and rightest cell.

**Figure 12 sensors-20-03877-f012:**
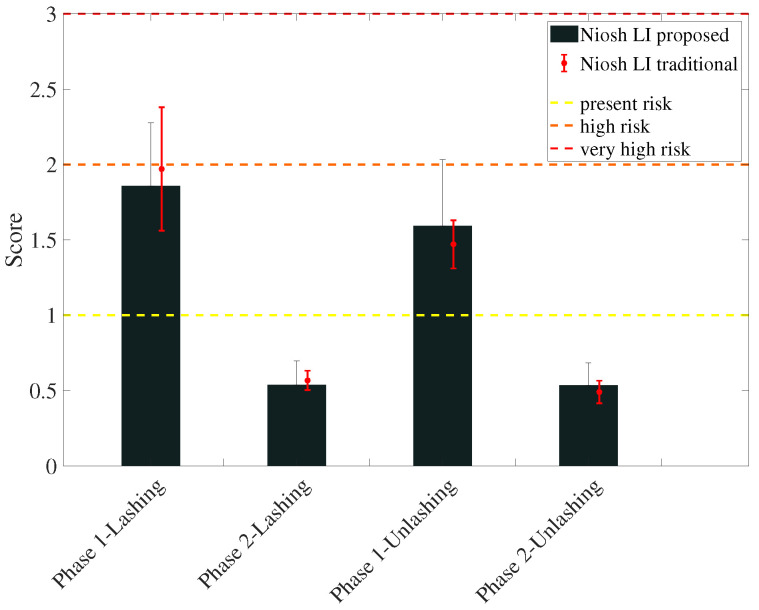
NIOSH scores for lasching and unlashing in phase 1 and phase 2. The black columns represent the average and the standard deviation of the NIOSH score with the proposed method. The average scores and their standard deviations obtained through the manual evaluation are represented in red. Three horizontal dashed lines represent the different risk thresholds: a possible risk in yellow, a high risk in orange and very high risk in red.

**Figure 13 sensors-20-03877-f013:**
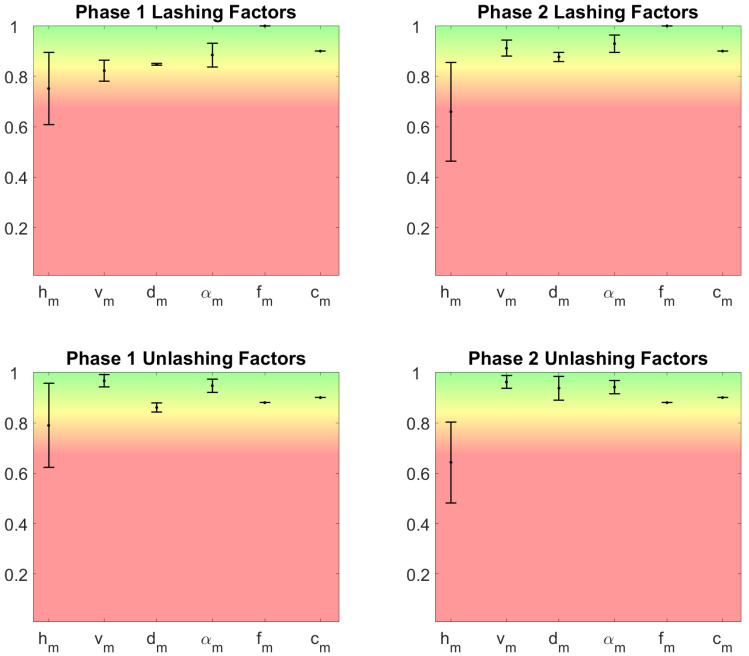
Factors that determines the Lifting Index in the two phases of Lashing and Unlashing operations. The background colors indicate the average factor values that, under the assumption $m_{a}=m_{ref}$, would produce a LI score corresponding to low (green), high (yellow t orange) and very high (red) risk.

**Figure 14 sensors-20-03877-f014:**
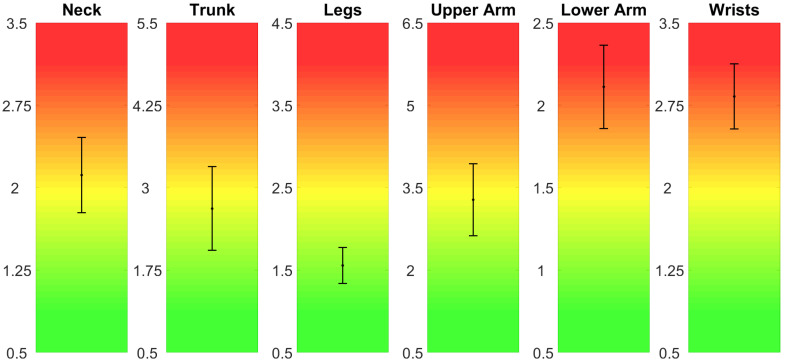
Distribution of scores for the postural factors of the REBA score. The background color varies between green (minimum value of the score) and red (maximum value of the score).

**Figure 15 sensors-20-03877-f015:**
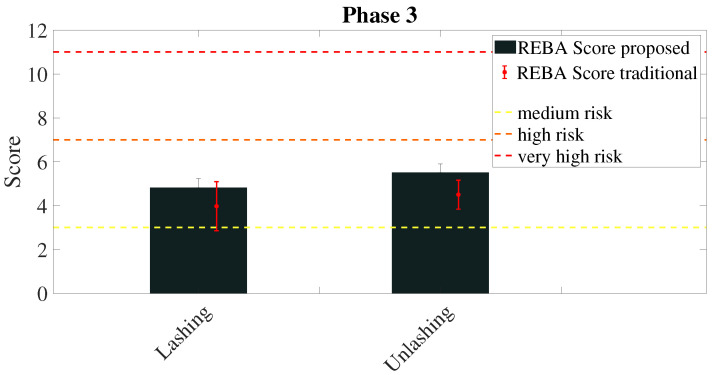
REBA score for lashing and unlashing in phase 3. The black columns represent the average and the standard deviation of the REBA score with the proposed method. The average scores and their standard deviations obtained through the manual evaluation are represented in red. The three horizontal dashed lines represent the different risk thresholds: a medium risk in yellow, a high risk in orange and very high risk in red.

**Figure 16 sensors-20-03877-f016:**
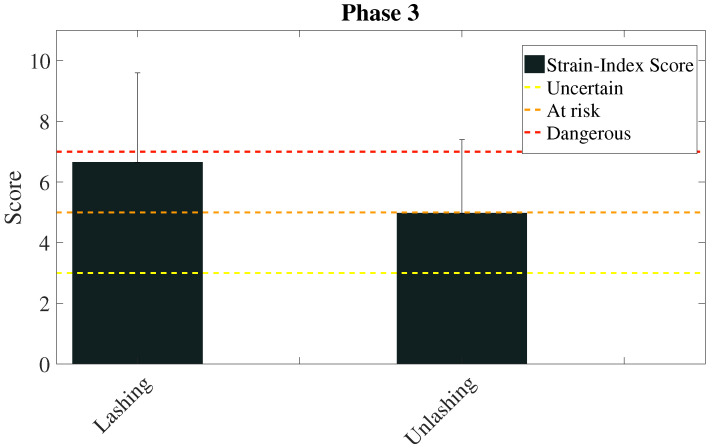
Strain Index score for lashing and unlashing in phase 3. The black columns represent the average and the standard deviation of the SI score with the proposed method. Three horizontal dashed lines represent the different risk thresholds: a low risk in yellow, a high risk in orange and very high risk in red.

**Table 1 sensors-20-03877-t001:** Characteristics of the three inertial systems presented: Xsens, Perception Neuron, and custom device.

Characteristics	Xsens	Perception Neuron	Custom
Suggested amount of IMUs	17	17	11
Transmission frequency [Hz]	240	120	100
Wireless protocol	Wi-Fi	Wi-Fi	Wi-Fi
Latency [ms]	20	10–13	3
Battery life [h]	9.5	3.5	5
Buffering [min]	10	-	-
Weight [kg]	1.13	0.85	0.54
Magnetic disturbances tolerant	yes	no	no

**Table 2 sensors-20-03877-t002:** Characteristics of the two EMG systems presented: Shimmer3 and custom.

Characteristics	Shimmer3	Custom
Suggested units number	2	2
Differential channels	4	4
Wireless protocol	Bluetooth	Wi-Fi
Transmission frequency [kHz]	0.125–8	0.250–32
Resolution [bit]	12	24
CMRR [dB]	-	−115
SNR [dB]	-	15–16 dB
SD card [GB]	8	-
Gain	1–12	1–12
Dimension [cm]	6.5 × 3.2 × 1.2	6 × 5 × 1.5 & 4 × 5 × 0.5
Latency [ms]	25	3
Battery life [h]	24	10
Unit weight [g]	45	110

**Table 3 sensors-20-03877-t003:** The NIOSH table represents the parameters that must be set manually and those calculated. Abbreviations: Y stands for YES, N stands for NO.

NIOSH		
	Task Analysis Input	Calculated
Age and Gender	Y	N
Lifted Mass	Y	N
Horz. Distance	N	Y (motion capture)
Vertical Location	N	Y (motion capture)
Vertical Displacement	N	Y (motion capture)
Asymmetry Angle	N	Y (motion capture)
Frequency	N	Y(motion capture)
Quality of the Grip	Y	N

**Table 4 sensors-20-03877-t004:** The Snook & Ciriello table represents the parameters that must be set manually and those calculated. Abbreviations: Y stands for YES, N stands for NO.

Snook & Ciriello		
	Task Analysis Input	Calculated
Gender	Y	N
The sustained and initial force	Y	N
Handle height	N	Y (motion capture)
Covered Distance	N	Y (motion capture)
Frequency	N	Y(motion capture)

**Table 5 sensors-20-03877-t005:** The REBA table represents the steps for REBA score evaluation and the parameters that must be set manually and those calculated. Abbreviations: Y stands for YES, N stands for NO.

REBA				
Step	Description	Variable	Task Analysis Input	Calculated
1	Neck Position	Head Flexion	N	Y (motion capture)
		Head is twisted	N	Y (motion capture)
		Head is side bending	N	Y (motion capture)
2	Trunk Position	Trunk Flexion	N	Y (motion capture)
		Trunk is twisted	N	Y (motion capture)
		Trunk is side bending	N	Y (motion capture)
3	Legs	Legs Flexion	N	Y (motion capture)
4	Posture Score A	Table A	N	Y (motion capture)
5	Load Score	Load Class	Y	N
		Shock or rapid build up of force	Y	N
6	Score A	Posture Score A + Load Score	N	Y (from table)
7	Upper Arm	Shoulder Flexion	N	Y (motion capture)
		Shoulder is raised	N	Y (motion capture)
		Upper Arm is abducted	N	Y (motion capture)
		Person is leaning	N	Y (motion capture)
8	Lower Arm Position	Elbow Flexion	N	Y (motion capture)
9	Wrist Position	Wrist Flexion	N	Y (motion capture)
		Wrist is bent	N	Y (motion capture)
		Wrist is twisted	N	Y (motion capture)
10	Posture Score B	Table B	N	Y (from table)
11	Add Coupling Score	Quality of the Grip	Y	N
12	Score C	Table C	N	Y (from table)
13	ctivity Score	Static position (longer than 1 min)	N	Y (motion capture )
		Repeated Action (more 4× per minute)	N	Y (motion capture)
		Action causes rapid large range changes	Y	N
Output	Final Score	Score C + Activity Score	N	Y (from table)

**Table 6 sensors-20-03877-t006:** The Strain Index table represents the parameters that must be set manually and those calculated.Abbreviations: Y stands for YES, N stands for NO.

Strain Index		
	Task Analysis Input	Calculated
Intensity of Exertion	N	Y (EMG)
Duration of Exertion	N	Y (EMG)
Exertions per minute	N	Y (EMG)
Wrist Posture	Y (for grip quality)	Y (motion capture for wrist posture)
Speed of Work	Y	N
Duration per Day	Y	N

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
