# Peer review of "Wearable Sensor Network for Biomechanical Overload Assessment in Manual Material Handling"

_sensors, 2020, doi:10.3390/s20143877_

Round 1

Reviewer 1 Report

REVIEW SENSORS 856021

Brief summary: The paper “Wearable Sensor Network for Biomechanical Overload Assessment in Manual Material Handling” presents a novel system for the evaluation of the ergonomic risk due to biomechanical overload capable of covering all areas of ISO 11228. The method advances the state of the art extending the procedure to the whole body, including the EMG for the evaluation of the effort and automatizing the score assessment with a time-saving process. It has been tested in Lift-on/Lift-off of containers port activity. The great novelty of this method is the semi-automatization of the calculation of the risk score, which therefore involves the evaluation of the factors used to calculate it, which can be investigated more in detail to decrease the risk associated with the work activity. However, the main weakness of the proposed method is the set-up of the neural network used for the activity segmentation, which may require time and the intervention of an expert if the MMH activity is particularly complex.

Broad comments: The topic of the present work is certainly original; the paper is well structured and interesting for the readers. Furthermore, the aim of improving the state of art about assessment of risk factors exposure in term of time-saving and with the addition of quantitative information by means of EMG signals is noticeable.

More in detail, the introduction draws readers' attention and the background section is very useful to understand the basis the project grows on. However, at the end of the section 2.2 it would be interesting to discuss the limits of direct measurement methods that constitute the state of the art, and how these have been solved by your system. (for example, the benefits that are achieved through the introduction of activity segmentation and the introduction of a total body system).

Materials and Methods section presents the architecture of the system and then develops on its various blocks in a superficial way. For instance, in Section 3.3, Task Analysis, it is explained deeply how TR12295 report has to be filled in general. Move this explanation to section 2.1, which is dedicated to the ergonomic assessment methods, and this general explanation results coherent with the description of those methods. Section 3.3, Task analysis, could be joined with Section 3.2 clarifying how the report is applied in your system, and especially what are the parameters that the rater enters manually with regard to this structure block.

In Section 3.4.1, speaking about the wearables systems, add the suggested positioning on the segments of the sensors (i.e. For Xsens MVN suit, you only cite segments number, 23, and sensors number, 17), and moreover, some example of HMM that could be studied mounting this sensors disposition. Also, it would be nice introducing a summary table with the characteristics, advantages and disadvantages of the three wearable inertial systems presented to show visually the differences.

 In Section 3.6, “joint variables as well as position and velocity relative to the chest of the ankles, the hands, and the hips” (lines 448-449) are mentioned as vector input of the MLP Neural Network. Explain how these variables can be obtained from IMUs raw data. If there is a process of integration starting from accelerations it is worth discussing the errors that these may entail and how they would influence the proposed risk score evaluation method.

In addition, the Section 3.7.2 regarding the offline analyses can be expanded, stressing out the strong point of the method introduced: the possibility of objectively checking which factors, starting from which the risk factor is calculated, are the ones that worsen the score. Once the aforementioned are known, the wrong worker’s attitudes related to these factors could be improved in the next work sessions.

Finally, Section 3.8 is more an estimation and a comparison of the computational time between the methods rather than a proper “comparison of the proposed method with a traditional one”. Change the title.

The section 4.1 of the case study describes impeccably the instruments, both software and hardware, adopted for the studied work activity; the calibration steps, the times needed for all the steps of the experiment, from preparation to data processing, are all indicated. However, there are still many gaps in the protocol. Figure 10 only shows visually the sensors worn, but a list of the segments where the inertial sensors have been placed is required. Moreover, as regards the segmentation process, it would be interesting to investigate what are the variables that define the 962 samples of the input vector of NN: which operations have been performed from motion data and EMGs RMS signals to find the relevant features used to feed the neural network?

The results are all well-presented through the figures in section 4.3, except for the labelling and segmentation process. Figure 11, in fact, presents undefined percentages, the equations that calculate them have not been explained. Even the numbers themselves within the confusion matrix seem to be wrong (So low percentages of correct classification?). The amount of effective work minutes that the system processes has been defined, which is a curious data compared to the total time of the activity. However, this data is not very relevant in relation to the neural network classification work; it would be much more valid to insert the activity numbers that characterize both training sets and test sets, which also have a strong statistical value.

In the discussion section, the analysis of the confusion matrix is totally absent, no comment has been made on the results obtained regarding the classification of the activities. On the other hand, the rest of the results are well investigated, supporting the hypotheses on which this project was based and validating the experiment in all its aspects through the comparison with the traditional method. Finally, the weaknesses of the method and the subsequent future work considered to refine the proposed method are also very clear, supporting the conclusion.

Regarding the figures, it seems to be some problems (Figure 7 is not cited), or they are in the wrong positions (Figure 5,8 & 9). Arrange them so that there is consistency between text and pictures.

English vocabulary and grammatical syntax could be improved in order to help readers in better understanding the paper content.

Specific comments:

  1. Line 2: “…in which an expert rater visually inspect videos of the working activity.” Change the verb “inspect” with “inspects”
  2. Line 27: “A large share of these disorders involves manual material handling (MMH).” must be justified or referenced.
  3. Line 41: change “to give the rater a score” with “to give to the rater a score”
  4. In lines 82-93 you list the parameters needed to calculate the score with the various methods. Why don’t you use a bulleted list and moreover add how these parameters can be obtained from IMUs and EMG data?
  5. Lines 116-118: “if the system provides the worker with information gives to the worker feedback concerning his current ergonomic behaviour, workers can correct their posture immediately.”
  6. Line 158: “The system is composed of a wearable component and of a software one”
  7. Line 188: what do you mean with device selection? The sensors could be used separately? How can you decide which sensors have to be on? Clarify this point.
  8. Line 205: you talk about “TR12295 report” stating you have mentioned it in section 2.1, although you cited it in Introduction.
  9. In Figure 4b, specify that a Shimmer3 single units of the whole device is worn by a volunteer.
  10. In section 3.4.3, latencies should be referenced (?).
  11. Line 345-346: “The highest overall latency is 100 ms, whereas the worst misalignment between motion and EMG data is 87 ms when the custom device and the Xsens suit are used together.” Should not it be 83ms? (maximum Xsens latency= 100ms; minimum EMG custom latency= 17ms -> 100-17= 83ms).
  12. Why isn’t Figure 5 positioned below Section 3.4.3?
  13. Could be useful adding a flowchart of the REBA score estimation with the 13 steps in Section 3.5.3?
  14. Line 457: “…the more data is have to be collected to set up and train the network.”
  15. In section 3.7, Figure 7 is not cited.
  16. Line 491: “In both the traditional and the proposed methods method,…”.
  17. Why isn’t Figure 8 positioned below Section 3.8?
  18. Why isn’t Figure 9 positioned below Section 4?
  19. Line 548: Add the comma. “As first step, the worker was informed about the research activity, the experiment is for and the usage and dissemination modalities of the acquired data.”.
  20. Lines 561-570: Add the accurate positioning of the inertial sensors on the subjects for this study case.
  21. Line 674: change “The method advances the state of the art as it include the EMG…” with “The method advances the state of the art as it includes the EMG…”
  22. Lines 691-692: change “…in addition to having objective measures of the motion and effort variables.” with “…in addition it leads to objective measures of the motion and effort variables.”

Author Response

The authors' reply to this reviewer's comments is included in the attached file.

The Authors

Reviewer 2 Report

The paper is well written. Authors need to improve conclusion section and also need to add some of the most recent papers as references.

Author Response

(The authors gave the same response as above.)

Reviewer 3 Report

This article proposes a system for the assessment of biomechanical overload, capable of covering all areas of ISO 11228.  It uses a sensor network composed of three IMU-based motion capture systems and two EMG capture devices. The proposed method showed consistency, time effectiveness, and potential for deeper analyses,  including intra-subject and inter-subjects variability, as well as a quantitative biomechanical analysis. The method advances the state of the art as it includes the EMG for the evaluation of the effort and it is extended to the whole body. The manuscript is largely well written.  Some comments to improve the quality of the manuscript.

Comments

  • “With the main goal of alleviating the burden for the rater.” Consider giving in the main goal a little more of importance to the welfare and health of the worker.  This type of technology not only alleviates the burden of the rater but facilitates the deployment of biomechanical overload risk assessment on MMH, potentially preventing WMSDs.
  • What type of electrodes were used, specifically for sEMG measurements?
  • Explain the cross-validation used in the neural network, when talking about precision: “Manual labeling stops when the achieved precision is at least 85% on average over the phases and the minimum precision is at least 80%.” Later on, you say that the data is split into 70-30%
  • Details on the internal neural network for activity recognition are missing. How long are the segments of data to be used by the NN?  Are there any default parameters, or the rater needs to set everything?  How accurate is it considering data from only one subject for training? Is it using data from other subjects to train?
  • Although some weaknesses of the method have been identified, consider further analyzing potential limitations of the system, for example considering the potential problems the system could encounter for deployment, the reduced dataset used for evaluation, technical details that could limit adoption of the technology by the users, etc.

Author Response

(The authors gave the same response as above.)

Reviewer 4 Report

In this work, the authors developed a wearable sensor network-based system for biomechanical overload assessment. The wearable system, which consists inertial measurement units (IMU) and EMG sensors, is capable of performing data capture, segmentation, and calculation of the ergonomic risk. Case study was performed to assess the performance of the proposed system and evaluate its effectiveness. The reported work is likely to advance the current state-of-the-art methods in the field. The research was welled designed and the clearly presented. This work can be accepted for publishing in Sensors after minor revision.

  • What's the total weight of the inertial systems and EMG system?
  • How do the wearables affect the work activities of the workers? Any tests and evaluation of the user experience?
  • How long can the EMG system run on the battery?
  • How does the EMG system deal with the motion induced artifacts?
  • The raw data from the inertial systems and EMG system in the case study should be provided to illustrate how the signals were processed and how the software help the rater with the data capture, segmentation, and calculation of the ergonomic risk.

Author Response

(The authors gave the same response as above.)
